# Counterfactual Sum-Product Networks

**Florian Peter Busch**[1,2]    **Moritz Willig**[1]    **Matej Zečević**[1]    **Kristian Kersting**[1,2,3,4]    **Devendra Singh Dhami**[5]

[1]Department of Computer Science, Technical University of Darmstadt, Darmstadt, Germany
[2]Hessian Center for AI (hessian.AI), Darmstadt, Germany
[3]Centre for Cognitive Science, Technical University of Darmstadt, Darmstadt, Germany
[4]German Center for Artificial Intelligence (DFKI), Darmstadt, Germany
[5]Department of Mathematics and Computer Science, Eindhoven University of Technology, Eindhoven, Netherlands

## Abstract

The complexity and vastness of our world can require large models with numerous variables. Unfortunately, coming up with a model that is both accurate and able to provide predictions in a reasonable amount of time can prove difficult. One possibility to help overcome such problems is sum-product networks (SPNs), probabilistic models with the ability to tractably perform inference in linear time. In this paper, we extend SPNs' capabilities to the field of causality and introduce counterfactual Sum-Product Networks (cf-SPNs), a type of SPNs capable of answering counterfactual questions. cf-SPNs make use of a neural component that sets the parameters of an SPN such that it represents the specified counterfactual world. We show that cf-SPNs can successfully learn counterfactual distributions.

## 1 INTRODUCTION

Consider the following example, which is an adaptation of a well-known scenario. Person $U$ keeps a small plant in their office but forgets to water it before going on a business trip. If person $U$ now remembers the plant ($U = 1$), a message $M$ is sent ($M = 1$) to two colleagues $A$ and $B$. Both colleagues water the plant ($A = 1$, $B = 1$) if they get a message, in which case the plant remains healthy ($H = 1$). In this example, a strong correlation between the plant being healthy and a message being sent can be observed, but it is clear that the plant's health has no causal impact on the message. To reason about causes and effects or to answer counterfactual questions such as "Given that the plant is healthy, would it still be healthy had $A$ not watered it?", an understanding of causality and its implications on the underlying structural equations is necessary.

Causality can be seen as the science centered around the study of causes and effects [Pearl, 2009, Bareinboim and

Pearl, 2016], which distinguishes between purely correlational observations and directed causal relations. Here, Pearl introduced the "ladder of causation" [Pearl and Mackenzie, 2018], which consists of an observational rung (correlations), an interventional rung (general causes and effects), and a counterfactual rung (hypothetical statements based on real-world evidence). Each further step on the causal ladder describes a more difficult problem that requires more information to solve. Starting from the second rung, models can differentiate between the plant's health *being correlated with* 'sending a message' and the *directed causal impact* of the message on the plant.

An example of probabilistic models that can reason causally is Causal Bayesian Networks (CBNs; Pearl [1995]). CBNs combine the advantages of Bayesian Networks, i.e., decomposing the joint probability distribution into a set of (interpretable) conditional distributions with the field of causality and can thus reach the interventional, second rung of the causal ladder. A major downside of CBNs is inference being intractable [Cooper, 1990]. While approaches exist that try to circumvent this problem using approximations [Murphy et al., 2013], it would be desirable to obtain causal models that can perform *exact inference in tractable time*. To achieve this objective, sum-product networks (SPNs) [Poon and Domingos, 2011] pose a promising alternative as they specifically allow for exact tractable inference.

Unfortunately, conventional SPNs only operate on the observational rung of the ladder of causation. With the goal of utilizing the tractable inference property of SPNs in the field of causality, we expand upon existing work of *interventional sum-product networks* [Zečević et al., 2021] and introduce counterfactual sum-product networks (cf-SPNs).

## 2 BACKGROUND AND RELATED WORK

Here, we explain the required background and give an overview of related work. We denote random variables by upper-case letters $V$, sets of random variables in boldface

*Accepted for the 8th Workshop on Tractable Probabilistic Modeling at UAI  (TPM 2025).*

**V**, values of single and sets of random variables as $v$ and $\mathbf{v}$ respectively, and probabilities of them as $P(v)$ or $P(\mathbf{v})$.

## 2.1 CAUSAL MODELS

In order to introduce the necessary background on causality, we follow Pearl's formalism of causal models and say that

**Definition 1.** *A structural causal model (SCM) is a tuple $\mathcal{M} \coloneqq \langle \mathbf{V}, \mathbf{U}, \mathbf{F}, P_{\mathbf{U}} \rangle$ over a set of variables $\mathbf{X} = \{X_1, \ldots, X_K\}$ taking values in $\boldsymbol{\mathcal{X}} = \prod_{k \in \{1 \ldots K\}} \mathcal{X}_k$ subject to a strict partial order $<_{\mathbf{X}}$, where*

- *$\mathbf{V} = \{X_1, \ldots, X_N\} \subseteq \mathbf{X}, N \leq K$ is the set of endogenous variables,*
- *$\mathbf{U} = \mathbf{X} \setminus \mathbf{V} = \{X_{N+1}, \ldots, X_K\}$ is the set of exogenous variables,*
- *$\mathbf{F} = \{f_1, \ldots, f_N\}$ is the set of deterministic structural equations, i.e. $V_i \coloneqq f_i(\mathbf{X}')$ for $V_i \in \mathbf{V}$ and $\mathbf{X}' \subseteq \{X_j \in \mathbf{X} | X_j <_{\mathbf{X}} V_i\}$,*
- *$P_{\mathbf{U}}$ is the probability distribution over the exogenous variables $\mathbf{U}$.*

The relationships between the variables as described by $\mathbf{F}$ induce the directed graph $G(\mathcal{M})$, which by definition is acyclic due to $<_{\mathbf{X}}$. The exogenous variables $\mathbf{U}$ are usually unobserved. We say that an SCM $\mathcal{M}$ entails the probability distribution $P_{\mathbf{V}}^{\mathcal{M}}$ over the set of endogenous variables $\mathbf{V}$.

Interventions in causal models change how a variable value is determined, ignoring what was previously defined in the set of functions $\mathbf{F}$.

**Definition 2.** *Consider an SCM $\mathcal{M} \coloneqq \langle \mathbf{V}, \mathbf{U}, \mathbf{F}, P_{\mathbf{U}} \rangle$ and a variable $V_i \in \mathbf{V}$. Applying an intervention $do(V_i = v_i) \in \mathcal{I}$ on $\mathcal{M}$ replaces the structural equation $f_i$ with $\tilde{f}_i \coloneqq v_i$ and results in the intervened SCM $\mathcal{M}_{do(V_i = v_i)} \coloneqq \langle \mathbf{V}, \mathbf{U}, \tilde{\mathbf{F}}, P_{\mathbf{U}} \rangle$ where $\tilde{\mathbf{F}} = (\mathbf{F} \setminus \{f_i\}) \cup \{\tilde{f}_i \coloneqq v_i\}$.*

A frequent assumption when using SCMs is the invariance of cause-effect relations (also known as invariance to the origin of the mechanism). Autonomy describes the invariance with respect to interventions, i.e., that conditional distributions of unintervened variables remain unchanged from interventions on different variables.

To extend our notion of SCMs to counterfactuals, we use the terminology of a "world" to describe a specific configuration of the entire set of endogenous variables.

**Definition 3.** *Consider an SCM $\mathcal{M} \coloneqq \langle \mathbf{V}, \mathbf{U}, \mathbf{F}, P_{\mathbf{U}} \rangle$, an original world $\mathbf{V}' = \mathbf{v}'$, and an intervention $do(V_i = v_i) \in \mathcal{I}$. Due to the (counterfactual) intervention, we have $\tilde{\mathbf{F}} = (\mathbf{F} \setminus \{f_i\}) \cup \{\tilde{f}_i \coloneqq v_i\}$. The distribution over the exogenous variables $P_{\mathbf{U}}$ is inferred to reproduce the original world $\mathbf{v}'$: $P_{\mathbf{U}}^{\mathbf{V}'=\mathbf{v}'} = P_{\mathbf{U}}(\mathbf{U}|\mathbf{V}' = \mathbf{v}')$. We call $\mathcal{M}_{do(V_i=v_i)}^{\mathbf{V}'=\mathbf{v}'} \coloneqq \langle \mathbf{V}, \mathbf{U}, \tilde{\mathbf{F}}, P_{\mathbf{U}}^{\mathbf{V}'=\mathbf{v}'} \rangle$ the counterfactual SCM.*

In SCMs, the entire randomness responsible for sample variability is captured by $P_{\mathbf{U}}$ since all functions computing $\mathbf{V}$ are deterministic. In other words, each sample $\mathbf{u}$ entails a specific setting of variables $\mathbf{v}$. Thus, given the original world $\mathbf{v}'$, it is possible to infer information about $\mathbf{u}'$.[1]

## 2.2 SUM-PRODUCT NETWORKS (SPNS)

An SPN is a probabilistic graphical model consisting of a directed acyclic graph (DAG) and a set of weights. Each leaf represents a probability distribution over a variable, and multiple leaves can correspond to the same variable but contain different probability distributions. The inner nodes are either sum or product nodes. In a product node, the child probability distributions are multiplied and in a sum node, a weighted sum over the children is calculated.

The following definition considers binary variables as a means of illustrating the concepts for computing probabilities with SPNs. An extension to continuous variables is without loss of generality made possible by having continuous distributions in the leaf nodes of the SPN, for further reference consider París et al. [2020]. Formally, we can describe an SPN $\mathcal{S} = (G, \mathbf{w})$ by a DAG [2] $G = (V, E)$ and the non-negative weights $\mathbf{w}$. Sum and product nodes are given by $\mathsf{S}(\boldsymbol{\lambda}) = \sum_{\mathsf{C} \in \mathsf{ch}(\mathsf{S})} \mathsf{w}_{\mathsf{S},\mathsf{C}} \mathsf{C}(\boldsymbol{\lambda})$ and $\mathsf{P}(\boldsymbol{\lambda}) = \prod_{\mathsf{C} \in \mathsf{ch}(\mathsf{P})} \mathsf{C}(\boldsymbol{\lambda})$, where $\boldsymbol{\lambda}$ is an indicator variable (IV). The SPN output is the value at the root node $\mathcal{S}(\boldsymbol{\lambda}) = \mathcal{S}(\mathbf{x})$ and probabilities can be computed by marginalization $P(\mathbf{x}) = \mathcal{S}(\mathbf{x}) / \sum_{\mathbf{x}' \in \mathcal{X}} \mathcal{S}(\mathbf{x}')$.

## 2.3 RELATED WORK

There are other models which compute counterfactual probabilities [Xia et al., 2023, Von Kügelgen et al., 2023, Bläser et al., 2025]. While Causal Bayesian Networks are powerful and can be transformed into SPNs and back [Zhao et al., 2015], this transformation from SPNs generally leads to degenerate[3] Bayesian Networks incapable of subsequent causal inference [Papantonis and Belle, 2020]. However, a model class extension, as used in this paper, poses a viable candidate for overcoming the problems of using SPNs for causal inference. Papantonisa and Bellea [2023] introduced an algorithm to transform SPNs into Bayesian Networks, which simplifies the calculation of interventional queries. Studying the complexity of counterfactuals, Han et al. [2022] highlighted that calculating counterfactuals using circuits is not any more complex than interventional or observational questions. Huber et al. [2023] consider counterfactuals in circuits by investigating partial identifiability.

---

[1] A step established as "abduction" in Pearl [2009].

[2] Not to be confused with a causal graph, which is also a DAG but not what is referred to here.

[3] A bipartite graph in which the actual variables of interest are not connected is called degenerate.

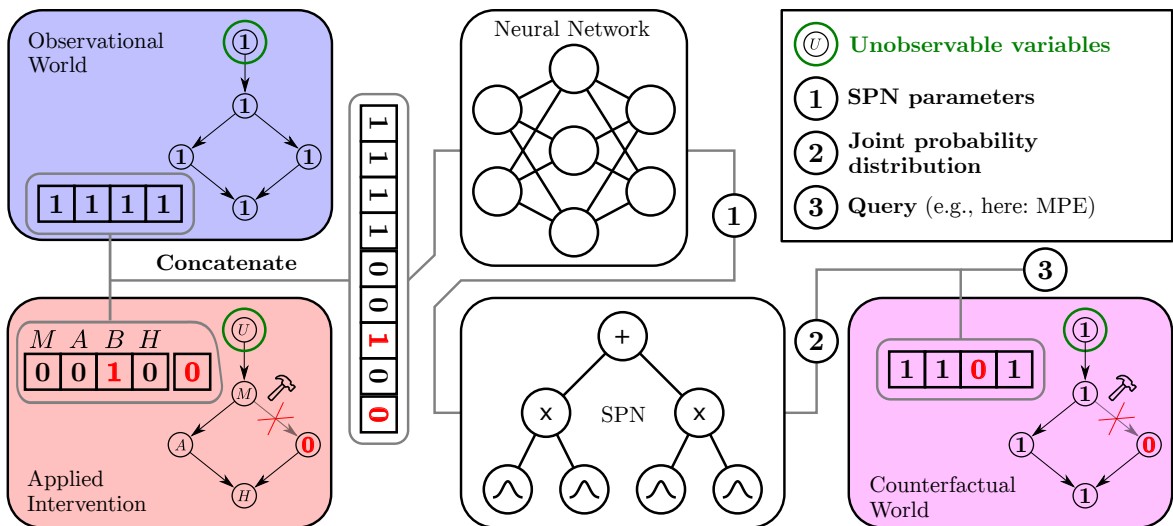

Figure 1: **Counterfactual SPN.** Computation of the counterfactual query $\arg\max_{\mathbf{X}} P(\mathbf{X}_{do(\neg B)}|M, A, B, H)$ using a cf-SPN. Information about the observational world (blue box) and the intervention (red box; indicating the intervened upon variables and their new value) is concatenated and given to a NN, which in turn computes the SPN parameters. The parameterized SPN can then be evaluated to answer counterfactual questions about a counterfactual world (violet box).

Recent work has used SPNs for generating counterfactuals for prediction tasks [Němeček et al., 2025], but they consider counterfactual predictions from an explainable artificial intelligence perspective and do not consider a wider range of counterfactuals for probabilistic problems. Zečević et al. [2021] introduced *interventional SPNs* (iSPNs) which use the same idea of a combined NN and SPN architecture to allow for interventional queries. Causal circuits have been shown to help with scaling [Busch et al., 2024].

## 3 COUNTERFACTUAL SUM-PRODUCT NETWORKS

Consider the following question based on the plant watering example: "Given that we know that person $B$ watered the plant, would the plant still be healthy had person $B$ not watered the plant?", i.e., $P(H_{do(\neg B)}|B)$. The query asks for the counterfactual value of $H$ under an intervention that sets $B$ to false, given that $B$ was true in the original world. To answer this, a model needs to infer the state of $A$ from the state of $B$, and incorporate the inferred knowledge about $A$ into the counterfactual world where $B$ is intervened.

We now propose the *Counterfactual Sum-Product Network* (cf-SPN), a tractable probabilistic model capable of answering counterfactual questions. We use an asterisk (*) to indicate variables of the counterfactual world.

**Definition 4.** *A* counterfactual sum-product network (cf-SPN) *is the joint model* $m(\mathbf{G}, \mathbf{D}) = g(\mathbf{D}^*; \boldsymbol{\psi} = f(\mathbf{D}', \mathbf{G}; \boldsymbol{\theta}))$, *where* $g(\cdot)$ *is an SPN,* $f(\cdot)$ *a non-parametric function approximator,* $\boldsymbol{\psi} = f(\mathbf{D}', \mathbf{G})$ *are shared parameters of the SPN,* $\mathbf{D}' \in \mathbb{R}^{K \times N}$ *and* $\mathbf{D}^* \in \mathbb{R}^{K \times N}$ *are data*

*matrices with observational and counterfactual values, respectively, and* $\mathbf{D} = (\mathbf{D}', \mathbf{D}^*)$.

$\mathbf{G} \in \{0, 1\}^{N \times N}$ is the (mutilated) causal graph according to some intervention $do(\mathbf{V}_j^* = \mathbf{v}_j^*)$. We use counterfactual data to train the model, such that the data matrix $\mathbf{D} \in \mathbb{R}^{K \times 2N}$ contains pairs of observational $\mathbf{D}' = \{\mathbf{V}_k'\}_k^K \in \mathbb{R}^{K \times N}$ and counterfactual $\mathbf{D}^* = \{\mathbf{V}_k^*\}_k^K \in \mathbb{R}^{K \times N}$ variable settings. This definition assumes complete evidence of both the observational and the counterfactual world.

Full computation for a cf-SPN is illustrated in Fig. 1. All original setting variables in the aforementioned example are set to true, and variable $B$ is intervened with a value of zero. Both vectors are concatenated and given to the NN, which outputs the parameters for the SPN $\boldsymbol{\psi}$. The resulting SPN estimates the distribution of the counterfactual world, such that all queries to the SPN are of a counterfactual nature. In our example, the desired probability for $H$ would be 1, indicating a 100% probability that the plant would still be healthy. This is correct as $A$ would still have watered the plant, even if $B$ would have been prevented from doing so.

**Proposition 1.** *Assuming autonomy and invariance, a cf-SPN* $m(\mathbf{G}, \mathbf{D})$ *is able to identify any counterfactual* $(\mathcal{L}_3)$ *distribution* $P^{\mathcal{M}}(\mathbf{V}_i^* = \mathbf{v}_i^* | \mathbf{V}' = \mathbf{v}', do(\mathbf{V}_j^* = \mathbf{v}_j^*))$, *permitted by a SCM* $\mathcal{M}$ *through counterfactuals, with knowledge of the mutilated graph* $G^*$, *the original world variables* $\mathbf{v}' \in \mathbf{D}'$ *generated from the original SCM, and corresponding counterfactual data* $\mathbf{v}^* \in \mathbf{D}^*$ *by modelling the distribution* $P^{\mathcal{M}^{\mathbf{V}'=\mathbf{v}'}_{do(V_i=v_i)}}(\mathbf{V}_i^* = \mathbf{v}_i^* | \mathbf{V}_j' = \mathbf{v}_j')$.

*Proof.* Let $\mathcal{M} := \langle \mathbf{V}, \mathbf{U}, \mathbf{F}, P_{\mathbf{U}} \rangle$ be the observational SCM. From the *do*-calculus [Pearl, 2009], we know that

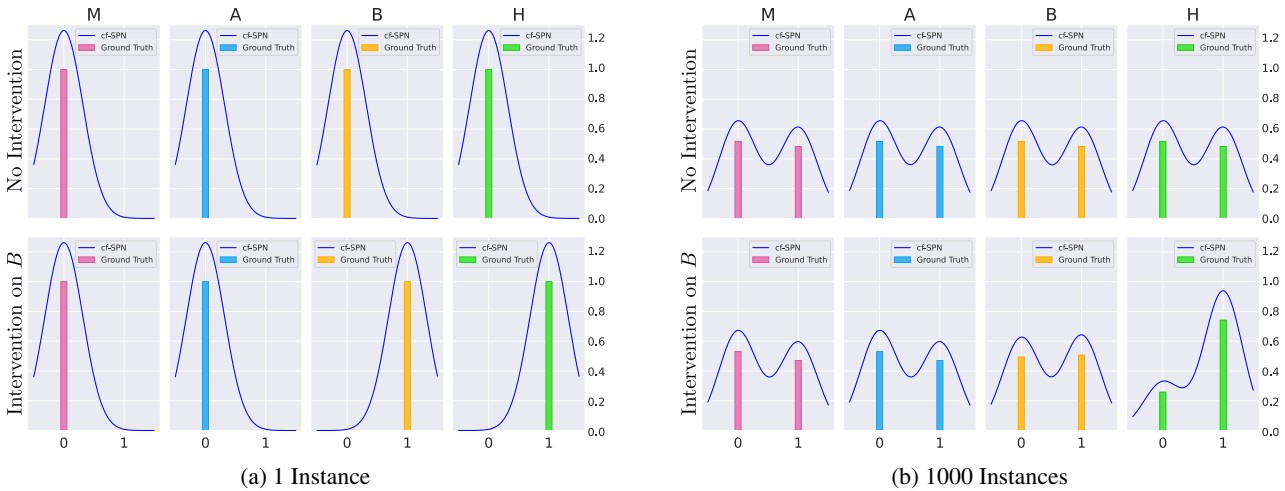

| (a) 1 Instance | (b) 1000 Instances |

Figure 2: **Watering Experiment.** cf-SPN predictions (blue lines) match the ground truth (bars) without any noticeable error. With information about the original world, counterfactual statements can be made reliably even about single instances.

$P^{\mathcal{M}}(\mathbf{V}_i^* = \mathbf{v}_i^* \mid do(\mathbf{V}_j^* = \mathbf{v}_j^*)) = P^{\mathcal{M}_{do(\mathbf{V}_j' = \mathbf{v}_j')}}(\mathbf{V}_i^* = \mathbf{v}_i^* \mid \mathbf{V}_j^* = \mathbf{v}_j^*)$. The counterfactual SCM $\mathcal{M}_{do(V_i = v_i)}^{\mathbf{V}' = \mathbf{v}'} := \langle \mathbf{V}, \mathbf{U}, \tilde{\mathbf{F}}, P_{\mathbf{U}}^{\mathbf{V}' = \mathbf{v}'} \rangle$ is equal to the interventional SCM $\mathcal{M}_{do(V_i = v_i)} := \langle \mathbf{V}, \mathbf{U}, \tilde{\mathbf{F}}, P_{\mathbf{U}} \rangle$ if $P_{\mathbf{U}}^{\mathbf{V}' = \mathbf{v}'} = P_{\mathbf{U}}$. For the specific sample $\mathbf{V}' = \mathbf{v}'$, we have $P_{\mathbf{U}}(\mathbf{U} \mid \mathbf{V}' = \mathbf{v}') = P_{\mathbf{U}}^{\mathbf{V}' = \mathbf{v}'}$. It remains to be shown that an SPN can learn the joint probability distribution $P(\mathbf{V}^*)$, which follows from Poon and Domingos [2011]. $\qquad\square$

Note that counterfactual data is required for training a cf-SPN. This is generally unobtainable, as it is generally impossible to directly "measure" values from the counterfactual outcome. This situation can be addressed in multiple ways. First, our model shows that SPNs are capable of answering counterfactual queries if the correct model is given, illustrating the potential of the proposed approach for domain-specific, expert-engineered models. Second, as shown in other works (e.g. Xia et al. [2023]), it is sometimes possible to calculate counterfactuals when only being provided with information from the lower rungs of the causal ladder.

## 4 EXPERIMENTS

We revisit the watering example to explain how the cf-SPN functions. The root variable $U$ is true $50\%$ of the time, and all other variables $M, A, B,$ and $H$ follow from it deterministically. The input to the NN of our cf-SPN consists of the intervention information (i.e., both the intervention target variable and the intervention value to be set) and a single configuration of the original world variables. The model is trained using counterfactual data by providing instances where the original world, the counterfactual world, and the intervention that distinguishes between them are known.

Figure 2 shows results for an intervention on variable $B$.

Because we use Gaussian leaves within the SPN, the model predicts continuous probability densities. We plot the densities in the range of $[-0.5, 1.5]$ (lines) while displaying the ground truth as discrete values (bars). For Figure 2a, all variables in the original world are set to false and –without an intervention– the counterfactual world is identical to the original world. When specifying an intervention on $B$, which sets it to 1, the cf-SPN infers that $H = 1$ is the correct counterfactual outcome. An intervention on $B$ indicates that person $B$ is set to water the plant, independent of getting the message $M$ or $A$ watering the plant. The predictions of the cf-SPN for $M$ or $A$ are therefore kept correctly unaltered.

Figure 2b follows the same type of setup but averaged over 1000 different original world instances with a $50\%$ chance of an intervention $do(B = 1)$. Without interventions, $U$ (and thus all other variables) would have been set to 0 and 1 half of the time. However, considering the average distribution over 1000 samples, $H$ is set to 1 more often. This is because the intervention on $B$ introduces a new possibility for $H$ being 1, namely the scenario where $U$ (and therefore $H$) would have been set to zero, but the counterfactual intervention on $B$ sets $B$, and therefore $H$, to 1.

## 5 CONCLUSION

We introduced cf-SPNs, tractable probabilistic models capable of computing counterfactual probabilities. We have shown that cf-SPNs are able to successfully make predictions for counterfactual questions. While our current model requires training on counterfactual data, one could try to utilize different information, for example, by including domain knowledge or counterfactual samples gained from human experts. Another challenge when applying these models is scalability and expressiveness for more difficult problems with a large number of variables.

## Acknowledgements

This work is supported by the Hessian Ministry of Higher Education, Research, Science and the Arts (HMWK; projects "The Third Wave of AI"). The authors acknowledge the support of the German Science Foundation (DFG) research grant "Tractable Neuro-Causal Models" (KE 1686/8-1). The Eindhoven University of Technology authors received support from their Department of Mathematics and Computer Science and the Eindhoven Artificial Intelligence Systems Institute.

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
