# OpenReview forum: "Counterfactual Sum-Product Networks"
_auai.org/UAI/2025/Workshop/TPM — TPM 2025_

### Official Review · Reviewer_Ewhi · 2025-06-11
**Novel idea potentially triggering interesting discussions (but the presentation could be improved)**

**Rating:** 2

**Review:**

The paper is an interesting attempt to apply the SPN formalism to causality following the direction drawn by Zecevic et al. (NeurIPS 2021) paper and moving from the interventional to the counterfactual ladder. This is an important direction for research in PCs, and the paper can trigger interesting discussions at TPM 2025.

The presentation can be improved. I understand that the 4-page limit does not help, but the impression I received while reading the paper is that of a long technical paper subject to multiple cuts to fit the limits imposed by the workshop. As a result, some technical details and formalisms are sometimes hard to follow. A more qualitative presentation would have been more effective.

Having available (and complete!) counterfactual data is a strong and somehow unrealistic requirement. The authors should discuss this point better, and the last paragraph of S3 is unclear.

Definition 4 is a key point of the paper. Yet, many important details are missing, even if Fig. 1 helps to understand the point.

I don't understand why the authors cannot consider adapting the notion of twin/counterfactual networks to SPNs to achieve similar results, probably with less data. The factual and the counterfactual models share many parameters, while in my understanding, the approximator function is now aware of that.

Autonomy and invariance should be defined.

The toy example used to explain counterfactuals might be ok, but I would use it to introduce them even from the beginning.

---

### Official Review · Reviewer_QDNM · 2025-06-14
**This is an interesting idea reminding me a hypernetwork**

**Rating:** 3

**Review:**

The paper aims at bringing the tractability to the world of counterfacturals. The idea, if I got it right, is to train a hypernetwork [1] to predict the parameters of an SPN for a given state of the world and the intervention. From the computation perspective, it make sense. It would be nice if authors referred to prior art [1], which has proposed a similar in the world of neural networks, as it seems to me extremely relevant.

I am not an expert here on causality and therefore I cannot judge, if the model make sense from the point of view of causality, i.e. if it does not violate any assumptions and constraints of that field. By a bystander point of (who have read few things about causality), it makes sense.

What I am worried more would be fitting the model to the real data. The experiment containing completely observed world is very unrealistic, but good for verifying the method.

[1] Ha, David, Andrew Dai, and Quoc V. Le. "Hypernetworks." arXiv preprint arXiv:1609.09106 (2016).